# Aqueous Organic Zinc-Ion Hybrid Supercapacitors Prepared by 3D Vertically Aligned Graphene-Polydopamine Composite Electrode

**DOI:** 10.3390/nano12030386

**Published:** 2022-01-25

**Authors:** Ruowei Cui, Zhenwang Zhang, Huijuan Zhang, Zhihong Tang, Yuhua Xue, Guangzhi Yang

**Affiliations:** School of Materials and Chemistry, University of Shanghai for Science and Technology, Shanghai 200093, China; crw2155@163.com (R.C.); 183762634@st.usst.edu.cn (Z.Z.); hjzhang@usst.edu.cn (H.Z.); zhtang@usst.edu.cn (Z.T.); xueyuhua@usst.edu.cn (Y.X.)

**Keywords:** zinc-ion supercapacitor, three-dimensional vertically aligned graphene, polydopamine, highly concentrated salt electrolyte

## Abstract

A three-dimensional vertical-aligned graphene-polydopamine electrode (PDA@3DVAG) composite with vertical channels and conductive network is prepared by a method of unidirectional freezing and subsequent self-polymerization. When the prepared PDA@3DVAG is constructed as the positive electrode of zinc-ion hybrid supercapacitors (ZHSCs), excellent electrochemical performances are obtained. Compared with the conventional electrolyte, PDA@3DVAG composite electrode in highly concentrated salt electrolyte exhibits better multiplicity performance (48.92% at a current density of 3 A g^−1^), wider voltage window (−0.8~0.8 V), better cycle performance with specific capacitance from 96.7 to 59.8 F g^−1^, and higher energy density (46.14 Wh kg^−1^).

## 1. Introduction

The construction and manufacture of electrochemical energy storage systems with high power and energy density, fast charge and discharge rates, and excellent cycle performance are of great importance to the rational use of energy [1,2,3]. Traditional energy storage devices mainly include batteries and supercapacitors, which are widely used in high energy density and high power ranges, respectively [4]. As energy storage components, batteries have the advantages of high energy density and facilitate long-term storage of electrical energy [5]; however, the disadvantages of low power density, low charge and discharge efficiency, poor cycle performance limit their applications [6,7,8]. Supercapacitors have the advantages of high capacity, high specific power density, fas t charge and discharge rates and excellent cycle performance, but the low energy density limits their application [9,10]. The electrochemical hybrid supercapacitors combine the advantages of capacitive cathode and battery type anode material and have high energy and power density together with excellent cycle performance [11].

A few organic materials are being studied as electrode materials for supercapacitors, such as polyaniline (PANI) [12,13], polypyrrole (PPy) [14,15], poly(3,4-ethylenedioxythiophene) (PEDOT) [16], and quinone-based organics [3,17]. In recent years, the quinone-based organic of polydopamine (PDA), which contains a large amount of catechol and nitrogenous amino groups, has been widely studied as electrocatalysts and electrode materials for energy conversion and storage. Wang et al. [18] constructed a fibrous aqueous zinc-ion battery with PDA, which has a large specific capacity (372.3 mAh g^−1^ at 50 mA g^−1^) and long-term cycle performance (80% capacity retention after 1700 cycles at 1 A g^−1^). While for the organic cathode, two factors hinder its development and application. Firstly, organic molecules are easily soluble in the electrolyte, resulting in low cycle performance. Secondly, most organics have poor electrical conductivity, which limits their multiplicative performance [19]. To solve the solubilization of quinone-based organic materials, many methods have been developed, such as separation modification, polymerization, mesoporous matrix constraint, and so on [20,21]. As for the problem of poor conductivity, studies have found that the ion/electron conductivity is closely related to the pore curvature of the electrode and the composite electrode, based on a highly conductive three-dimensional graphene structure, which can effectively improve the multiplicative performance [22]. In particular, three-dimensional vertically oriented graphene (3DVAG) has a special structure with vertically open channels and low pore curvature, which contributes to fast ion/electron transfer and, therefore, excellent performance.

In this work, unidirectional freezing and thermal reduction are used to prepare 3DVAG with a 3D long-range ordered structure. Three-dimensional vertically aligned graphene-polydopamine (PDA@3DVAG) composite electrodes were prepared by loading PDA particles on the as-prepared 3DVAG substrates via oxidative self-polymerization. By constructing PDA@3DVAG as the positive electrode of zinc-ion hybrid supercapacitors (ZHSCs), excellent electrochemical performances are obtained in highly concentrated salt electrolytes for both three and two electrode systems.

## 2. Experimental

Figure 1 illustrates the preparation process of PDA@3DVAG electrode material. In brief, 3DVAG aerogel was prepared by directional freezing and subsequent thermal reduction. Firstly, LGO suspension (2 mL, 5 mg mL^−1^) was mixed with ascorbic acid (VC, 20 mg), which was subsequently heated in an oil bath for the first hydrothermal reduction to obtain partially reduced graphene oxide (PrGO) hydrogels. Then, PrGO was further heated in a water bath, and impurities were removed by deionized water. Finally, after freeze-drying and reduction, 3DVAG was obtained. Subsequently, PDA@3DVAG was prepared by the liquid-phase graphene reduction and oxidative self-polymerization process. Finally, it was assembled into a buckled supercapacitor for electrode testing.

### 2.1. Preparation of 3DVAG

Preparation of PDA@3DVAG 3D vertically oriented graphene was prepared by hydrothermal-assisted unidirectional freezing and subsequent thermal reduction. Firstly, 2 mL solution of graphene oxides suspension (5 mg mL^−1^) and 20 mg of ascorbic acid were mixed and heated in an oil bath for the first hydrothermal reduction to obtain partially reduced graphene oxide (PrGO) hydrogel. Then, the PrGO was placed on the surface of a copper ingot impregnated with liquid nitrogen for 5 min of unidirectional freezing. After thawing at room temperature, PrGO was reduced in a water bath for 6 h. The obtained gel was washed with deionized water to remove soluble impurities. After being chemically reduced by 20 μL 80 wt.% of hydrazine hydrate and subsequently washing, 3DVAG was finally formed.

### 2.2. Preparation of PDA@3DVAG

PDA@3DVAG composite electrode material was prepared by liquid-phase graphene reduction and one-step oxidative self-polymerization. Firstly, 0.15 g dopamine hydrochloride was dissolved in 75 mL water, then the pH of the dopamine hydrochloride solution was adjusted to 8.5 by adding trimethyl methylamine. Subsequently, 3DVAG hydrogel was impregnated into the solution to self-polymerize for 24 h. Finally, PDA@3DVAG was obtained after washing and freeze-drying.

### 2.3. Materials and Electrochemical Characterization

The morphologies of the samples were characterized using field emission scanning electron microscopy (SEM, Dutch FEI) and energy dispersive spectrometer (EDS, Dutch FEI) mapping scanning. The structure of PDA@3DVAG was characterized by Fourier transform infrared (FTIR, Perkin Elmer Spectrum 100) spectroscopy spectra and X-ray diffraction (XRD, Bruker D8-Advanced diffractometer). UV–vis spectroscopy was performed on electrolyte solutions with a Perkin Elmer LAMBDA 750.

The cyclic voltammetry (CV) and galvanostatic charge–discharge (GCD) of the as-prepared PDA@3DVAG composite electrode materials were measured via a three-electrode system in this work. In 2M ZnSO_4_ aqueous electrolyte, the electrochemical performances were tested with PDA@3DVAG composite electrode as working electrode, platinum sheet as counter electrode and Ag/AgCl electrode as reference electrode under the voltage window of 0~0.7 V. PDA@3DVAG composites were assembled into buckle supercapacitor for two-electrode testing. Porous activated carbon (AC) was used as the anode electrode, and CV tests were performed under the voltage window of −0.8~0 V.

The CV was performed by applying a linear voltage between the upper and lower limits of the two voltages for the working electrodes to perform a cyclic scan. The GCD examined the electrochemical response to controlled current, which is the response voltage versus time curve obtained by controlling a constant current.

The mass specific capacitance (*C_m_*, F g^−1^) was calculated as follows:(1)$$C_{m}=\frac{I\times \Delta t}{\Delta U\times m}\;\;$$
where *I* was the constant discharge current (A), Δ*t* was the discharge time (s), Δ*U* was the discharge voltage window (V), and *m* was the mass of active substance (g).

The mass energy density (*E_m_*, Wh kg^−1^) was calculated as follows:(2)$$E_{m}=\frac{1}{2}\times \frac{C_{m}\times {\left(\Delta U\right)}^{2}}{3.6}$$

The mass power density (*P_m_*, W kg^−1^) was calculated as follows:(3)$$P_{m}=3600\times \frac{E_{m}}{\Delta t}$$

## 3. Results and Discussion

### 3.1. Morphology and Structure of PDA@3DVAG

Figure 2 shows the SEM images and EDS analysis of PDA@3DVAG. The ordered porous structure can be observed in Figure 2a, where the channels show vertical orientation with a pore size of about 20–30 μm. Figure 2b,c shows the partial enlargement, where self-polymerized PDA particles are uniformly loaded on graphene substrates. As shown in Figure 2d, the EDS spectrum of PDA@3DVAG shows the main elements are C, N, and O. The mass fraction of C element occupies 91.43%, while N and O elements occupy 3.2% and 5.37%, respectively. Figure 2c,d indicate the loading amount of PDA is small and a thin layer is coated on the surface of graphene, which ensures a good contact between PDA particles and graphene surfaces.

Figure 3a illustrates the FTIR spectra of 3DVAG and PDA@3DVAG. Compared with 3DVAG, more functional group spectral peaks can be observed in the spectrum of PDA@3DVAG. The absorption peaks located at 3394, 3227, and 1721 cm^−1^ are found existing in the spectrum of PDA@3DVAG, which are induced by the stretching vibrations of the C−OH, N−H, and C=O group, respectively. The broad absorption peaks located at 1000~1800 cm^−1^ are induced by the vibrations of the indole group. The peaks located at 1609 and 1517 cm^−1^ are induced by the stretching vibrations of C=C on the benzene ring and C=N in the carbon−nitrogen five−membered ring, respectively [18,23]. The above results indicate that PDA is obtained by situ polymerization on the surface of graphene to form a PDA@3DVAG composite. As shown in Figure 3b, the XRD pattern of PDA@3DVAG reveals the characteristic peak of graphene structure (2θ = 26.1°), indicating a composite of graphene, which is consistent with FTIR.

### 3.2. Electrochemical Performance

Figure 4a shows the CV curve of the composite electrode at the scan rates as 2~100 mV s^−1^, from which a pair of redox peaks can be clearly observed. The forward CV scan shows a broad oxidation peak with a peak potential of ~0.46 V vs. Ag/AgCl. In this experiment, Ag/AgCl is used as the reference electrode, and its potential is 0.2 V. The coordination potential of the quinone group with Zn ion is 0.46 V vs. Ag/AgCl by conversion. It is consistent with the potential of the oxidation peak in the CV curve, which is the current response of coordination reaction between the quinone group in PDA and Zn ion. A reduction peak appearing at ~0.26 V vs. Ag/AgCl has a potential difference of 0.2 V from the oxidation peak, which is consistent with the literature (the reduction potential is ~3.1 V vs. Li) [19]. The reduction peak reflects the reduction in the benzoquinone to a catechol group, resulting in the loss of electrochemical activity and the occurrence of Zn^2+^ ligand detachment. The above analysis shows PDA has a reversible Faradaic reaction process of Zn^2+^ embedding and de−embedding. Figure 4b shows the GCD curves of the composite electrode at current densities of 0.5~2 A g^−1^. As the current density increases, the mass specific capacitance decreases. The highest specific capacitance is 142.4 F g^−1^ at the current density of 0.5 A g^−1^.

As shown in Figure 5a,b, the CV of the AC anode is performed under the voltage window of −0.8~0 V. The CV curve is rectangular−like, indicating that the anode stores zinc ions with electrostatic double−layer behavior. The PDA@3DVAG exhibits the apparent redox peaks in the CV curve and is a battery type as a cathode. The two electrode materials are assembled to construct an organic zinc−ion hybrid supercapacitor for further investigation of performances.

Figure 5c shows the CV curves of the ZHSCs at the scan rate of 20 mV s^−1^ for a voltage window of −0.8~0.7 V. Two pairs of redox peaks can be clearly observed in Figure 5c, one oxidation peak appears at 0.26 V, and another reduction peak appears at −0.2 V. The potential difference is 0.46 V, which is consistent with that (1.44 − 0.98 = 0.46 V) of the redox peak (Figure 5d) in the Zn//PDA@3DVAG zinc−ion battery (ZIBs). In the two sets of CV curves, the relative positions of the two pairs of redox peaks are close and their shapes and sizes are similar, indicating that the energy storage effect of the AC anode as the anode electrode is similar to that of metal zinc, while the mechanism is different. A reversible dissolution/deposition process occurs for metal zinc anode, while an electrostatic adsorption behavior for AC anode. Because of the low oxidation potential of AC anode and low redox potential of PDA, the assembled ZHSCs exhibit a low potential of −0.8~0.7 V, while the zinc cell voltage window can reach 0.5~1.7 V. However, the CV area of ZHSCs is larger than that of Zn−ion batteries, and the redox peaks at both locations are remarkable, indicating that the PDA is more electrochemically active in ZHSCs and has a greater ability to bind zinc ions.

The PDA@3DVAG electrode is impregnated in a 2M ZnSO_4_ electrolyte to observe its solubility. There is a color change from colorless and transparent to slightly brown after 24 h, as shown in Figure 6a. UV–vis spectroscopy of the electrolyte solution (Figure 6b) reveals an absorption peak at 281 nm obtained after 24 h of impregnation, which is consistent with the absorption peak of dopamine, indicating the presence of the soluble dopamine group in the conventional 2M ZnSO_4_ solution [18]. In order to inhibit the dissolution of active substances, an ultra-high concentration zinc salt electrolyte (18 m ZnCl_2_ + 6 m NH_4_Cl, WIS) is used. As shown in Figure 6a, there is no obvious color change, and the UV–vis spectra can also reveal no absorption peak of the dopamine group at 281 nm, indicating that the highly concentrated salt electrolyte can significantly inhibit the dissolution of PDA.

The electrochemical performances of PDA@3DVAG are compared in the two aqueous electrolytes of 2M ZnSO_4_ and highly concentrated salt. Figure 7a,c shows the CV curves of ZHSCs constructed with the two electrolytes at different scan rates from 2 to 100 mV s^−1^. As shown in Figure 7a, the PDA@3DVAG//2M ZnSO_4_//AC ZHSCs shows two pairs of redox peaks under the −0.8~0.6 V voltage window, with oxidation peak potentials of ~−0.1 V and ~0.3 V, and reduction peak potentials of ~0.1 V and ~−0.3 V, corresponding to the coordination reaction of Zn^2+^ and H^+^ with quinone group in the PDA, respectively. The highly concentrated salt electrolyte exhibits a wider voltage window (−0.8~0.8 V), as shown in Figure 7c. Due to the introduction of NH_4_^+^ in the electrolyte, the redox peaks in the CV curves of PDA@3DVAG//18 m ZnCl_2_ + 6 m NH_4_Cl//AC ZHSCs are −0.3 V and 0.35 V, and the reduction peak potentials are 0.15 V and −0.65 V, corresponding to the coordination reaction of Zn^2+^ and NH_4_^+^ with quinone group, respectively. According to the potential difference between AC anode and metal zinc, the redox peaks at 0.35 V and 0.15 V are coordination reactions occurring in Zn^2+^, which is consistent with the three−electrode system.

Figure 7b,d shows the GCD curves of the two ZHSCs at different current densities. As shown in Figure 7b, the symmetrical GCD curves can be observed in the conventional electrolyte ZHSCs, which indicates the good electrochemical performance of ZHSCs in conventional electrolytes. As the current density increases from 0.5 to 3 A g^−1^, the specific capacitance decreases from 128 F g^−1^ to 71.1 F g^−1^, with the capacitance retention of 55.54%. In the GCD curves of the highly concentrated salt electrolyte ZHSCs (Figure 7d), two pairs of charge and discharge plateaus can be clearly observed, corresponding to the redox peaks in CV. As the current density increases from 0.5 to 3 A g^−1^, the specific capacitance decreases from 133.9 F g^−1^ to 65.5 F g^−1^, with the capacitance retention of 48.92%. The specific capacitances of the two electrode materials at different current densities are also shown in Figure 7e. It can also be seen that the PDA@3DVAG with vertical orientation exhibits better multiplier performance compared to the system of PDA//WIS//AC ZHSCs with the capacitance retention of 40.67%. This result can also be found in the electrochemical impedance spectrum (EIS), as shown in Figure 7f. The slope is higher in the AC−based electrode, which indicates double−layer capacitance and not diffusion, while the PDA/WIS/AC follows a diffusion pattern. Therefore, there is a fundamental difference between these systems. In the Nyquist plot, the semicircular arc appearing in the high−frequency range is the charge transfer resistance, which is mainly affected by the contact interface resistance between the electrolyte and the electrode. The sloping line in the low-frequency range reflects the ion transfer resistance of the pores inside the electrode. Both PDA@3DVAG//2M ZnSO_4_//AC ZHSCs and PDA@3DVAG//18 m ZnCl_2_ + 6 m NH_4_Cl//AC ZHSCs have higher slopes in the low−frequency range due to the presence of carbon material in the positive electrode, but there is still ion diffusion resistance due to the presence of PDA pseudocapacitive material. The main energy storage contribution of PDA//WIS//AC is pseudocapacitance, so it exhibits large ion diffusion resistance. As shown in the enlarged figure of the high−frequency range, PDA@3DVAG exhibits a smaller ionic conductivity in WIS electrolytes compared to that in 2M ZnSO_4_, mainly due to the low ionic conductivity of the highly concentrated electrolyte. However, it has a positive effect on pseudocapacitance, which has a priority of high energy density compared with traditional Electrochemical Double−Layer Capacitors (EDLCs). The EIS characterizations of the two electrolytes exhibit different ion transport resistances, with the conventional electrolyte having a smaller equivalent resistance than the highly concentrated salt electrolyte. In the low−frequency range, conventional electrolytes have a higher slope than highly concentrated electrolytes, indicating a lower resistance to ion transport, while PDA electrodes exhibit the poorest ion diffusion. From the three−dimensional vertical channels and the conductive network, PDA@3DVAG ensures good ion diffusion, achieves excellent multiplicity performance under highly concentrated electrolytes, and improves the voltage window and achieves better electrochemical performance [24,25].

Figure 8 shows the cycle performances of conventional and highly concentrated salt electrolytes for ZHSCs at the current density of 1 A g^−1^. After 3000 cycles, the specific capacitance of the conventional electrolyte decreases from 95.8 to 42.3 F g^−1^, with poor capacitance retention of only 44.2%, which may be caused by the dissolution of PDA during the long-term charge and discharge process as discussed before. The highly concentrated salt electrolyte shows a decrease in specific capacitance from 96.7 to 59.8 F g^−1^ after the long-term cycles, with a relatively high capacitance retention rate of 61.8%, which is improved by 17.6% compared with that of the conventional electrolyte.

Figure 9 shows the Ragone plot of ZHSCs constructed with highly concentrated salt electrolytes. It can be found that the PDA@3DVAG composite electrode shows both high energy and power densities. The energy density reaches 46.14 Wh kg^−1^ at the power density of 393.75 W kg^−1^ and 19.29 Wh kg^−1^ at 2183 W kg^−1^, respectively. The energy density exceeds that of conventional electrochemical capacitors and some large−size batteries. As shown in Table 1, compared with the ZHSCs reported in the literature, the ZHSCs constructed with PDA@3DVAG composite electrodes exhibit an excellent energy density and good power density.

## 4. Conclusions

The selection of material and the design of structure for electrode are of great importance for the performances of ZHSCs, which requires large surface area, good electric conductivity, certain channels with a proper diameter of pores, and so on for ion and mass transfer, abundant active chemical sites, etc. In this work, a method of unidirectional freezing and subsequent self-polymerization is used to obtain a PDA@3DVAG composite, which has a three-dimensional vertically aligned structure with long ordered channels and uniform pores. When the composite is assembled as electrode materials for an aqueous organic zinc-ion hybrid supercapacitor, the electrochemical performances are fairly well with a wide voltage window, good cycle performance, and high energy density, especially in highly concentrated salt electrolyte systems. The structure of three-dimensional vertically aligned graphene and the composite of PDA polymer may provide some references for the development of high-performance aqueous organic zinc-ion energy storage devices.

## Figures and Tables

**Figure 1 nanomaterials-12-00386-f001:**
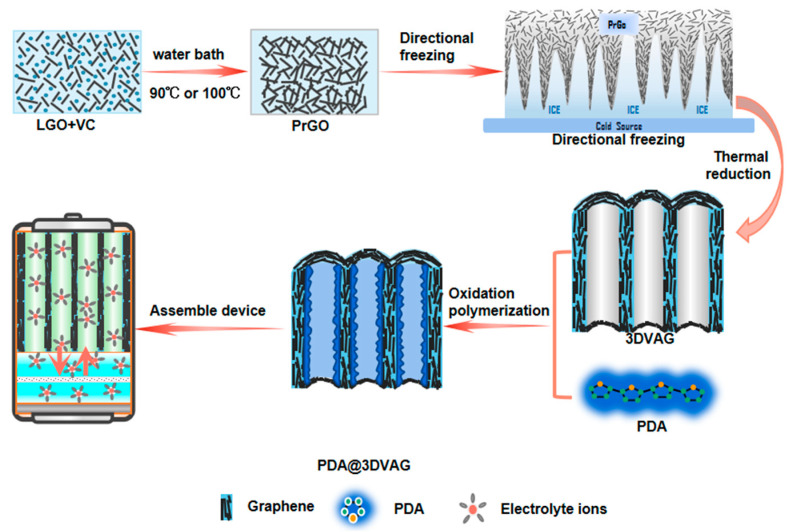
Illustration of preparation process of PDA@3DVAG composite electrode material.

**Figure 2 nanomaterials-12-00386-f002:**
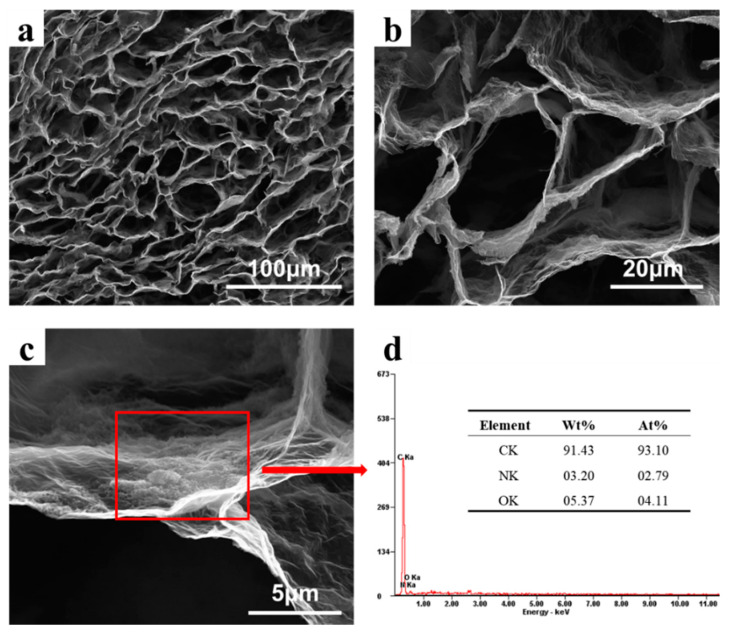
SEM images (**a**,**b**) and EDS analysis (**c**,**d**) of PDA@3DVAG.

**Figure 3 nanomaterials-12-00386-f003:**
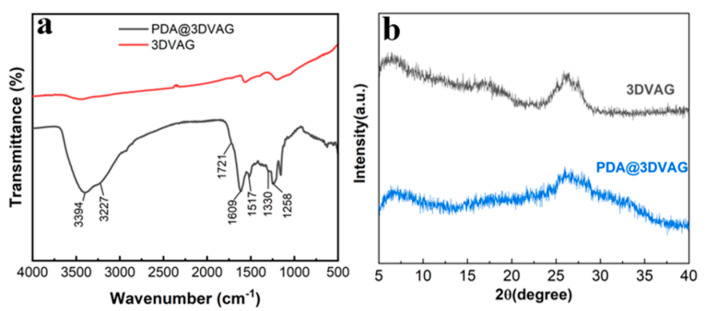
Chemical composition and structural characterization of PDA@3DVAG: (**a**) FTIR; (**b**) XRD.

**Figure 4 nanomaterials-12-00386-f004:**
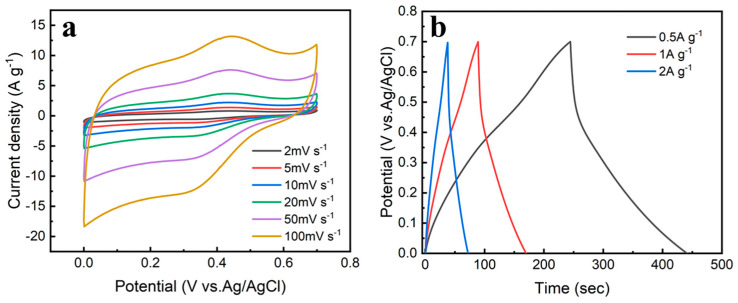
Electrochemical performances of PDA@3DVAG composites: (**a**) CV curves at different scan rates; (**b**) GCD curves at different current densities.

**Figure 5 nanomaterials-12-00386-f005:**
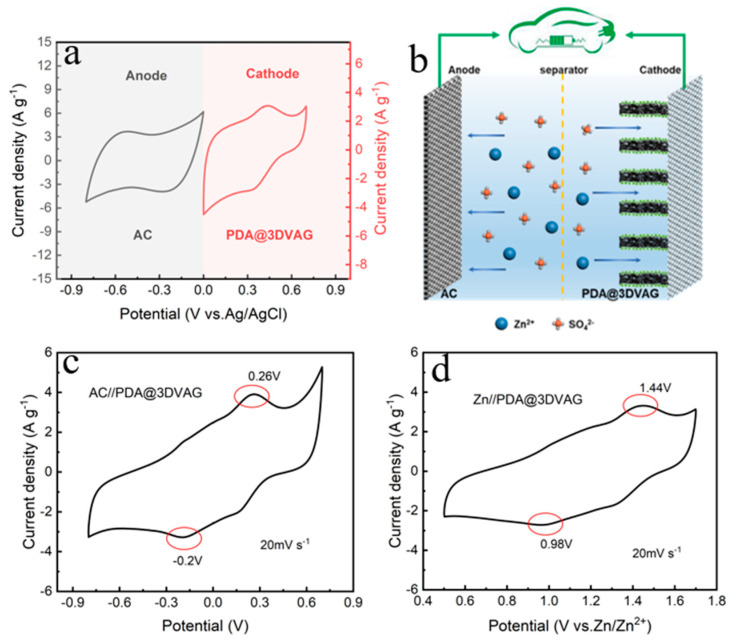
(**a**) CV curves of AC negative and PDA@3DVAG positive at a scan rate of 20 mV s^−1^; (**b**) Schematic diagram of ZHSCs structure; (**c**) CV curves of AC//PDA@3DVAG ZHSCs; (**d**) CV curves of Zn//PDA@3DVAG ZIBs.

**Figure 6 nanomaterials-12-00386-f006:**
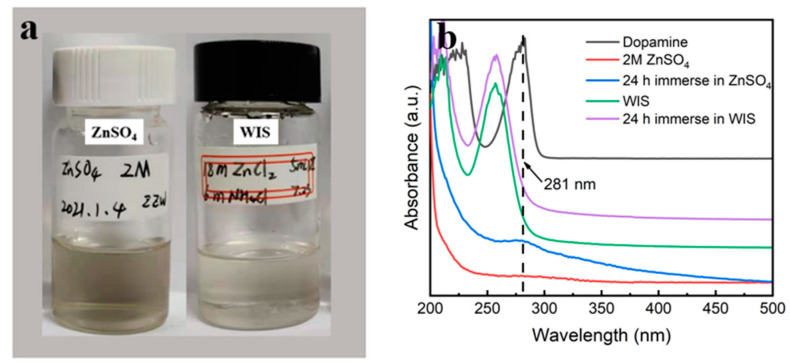
Solubility of PDA@3DVAG composite in two electrolytes: (**a**) camera picture; (**b**) UV–vis spectra.

**Figure 7 nanomaterials-12-00386-f007:**
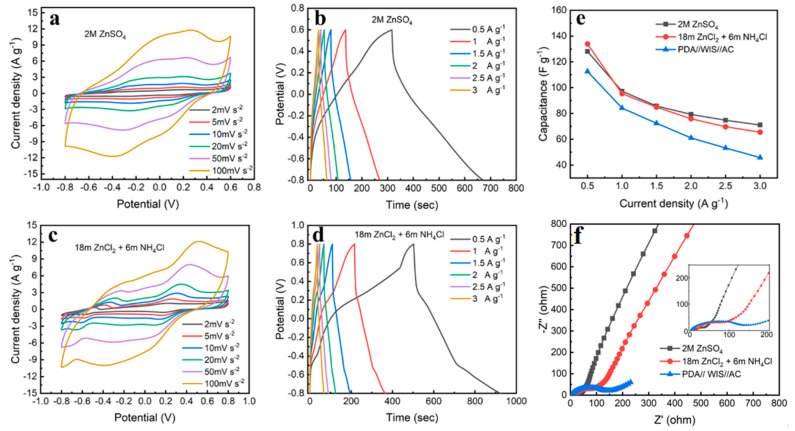
Electrochemical performances of evaluation: (**a**,**b**) CV and GCD curves of 2M ZnSO_4_ ZHSCs; (**c**,**d**) CV and GCD curves of highly concentrated salt electrolytes ZHSCs; (**e**) multiplicative performance plots at different current densities; (**f**) electrochemical impedance spectra (enlarged figure of high−frequency range exhibited in inset).

**Figure 8 nanomaterials-12-00386-f008:**
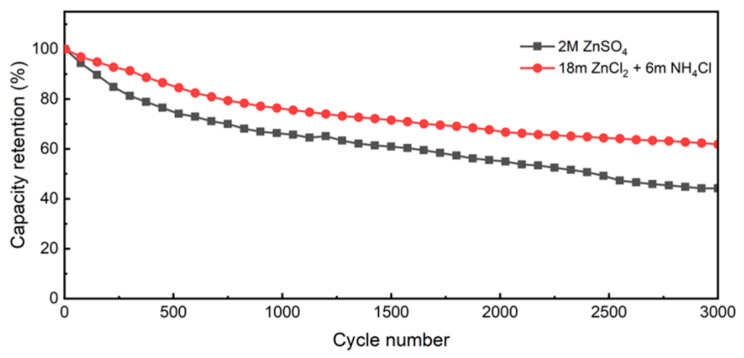
Cycle performances of different electrolytes for ZHSCs at the current density of 1 A g^−1.^

**Figure 9 nanomaterials-12-00386-f009:**
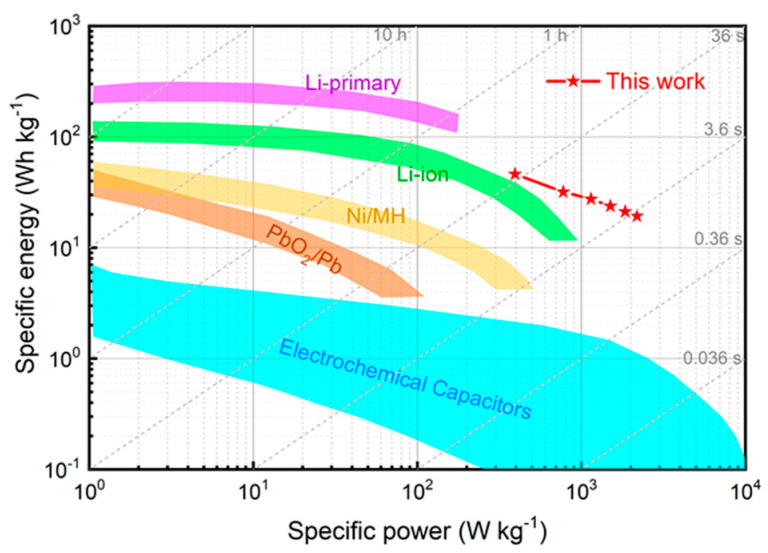
Ragone plot of ZHSCs constructed with highly concentrated salt electrolytes.

**Table 1 nanomaterials-12-00386-t001:** Performance of ZHSCs constructed by PDA@3DVAG composites compared with other systems.

Electrode Material	Electrolyte	Voltage (V)	Energy Density (Wh kg^−1^)	Power Density (W kg^−1^)	Reference
PDA@3DVAG//AC	WIS	0.8	46.14	393.75	This work
NTC	1M H_2_SO_4_	0.8	4.5	40	[26]
V_2_O_5_//AC	2M ZnSO_4_	2	34.6	1300	[27]
MnO_2_–CNTs//MXene	2M ZnSO_4_	1.9	29.7	2480	[28]
V_2_O_5_–ECF//ECF	6M LiCl	2	22.3	1500	[29]
LiNi_0.5_Mn_1.5_O_4_//AC	1M LiPF_6_	1.75	19	103	[30]
TiO_2_@EEG//EEG	1M LiPF_6_	1.5	10	2000	[31]
rGO/COF//rGO	1M H_2_SO_4_	1	10.3	50	[32]

## Data Availability

The data presented in this study are available from the corresponding author, upon reasonable request.

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
