# Peer review of "Aqueous Organic Zinc-Ion Hybrid Supercapacitors Prepared by 3D Vertically Aligned Graphene-Polydopamine Composite Electrode"

_nanomaterials, 2022, doi:10.3390/nano12030386_

Round 1

Reviewer 1 Report

The authors show an interesting method of preparing graphene-polydopamine electrodes (PDA@3DVAG) consisting of unidirectional freezing and subsequent self-polymerization resulting in three-dimensional channels to be used as cathodes in ZHSCs giving an acceptable electrochemical response, as seen in comparison with other similar systems. The article is well developed and argued. However I miss in this work a characterization of the microstructure of the resulting electrode, in particular XRD and an analysis of the pore size by superfie BET of the material. Can the authors specify a little better the microstructure of these systems?

Reviewer 2 Report

Yang and co-workers reported a PDA@3DVAG electrode with vertical channels and conductive network by unidirectional freezing and subsequent self-polymerization method for zinc-ion hybrid supercapacitor. In a highly concentrated salt electrolyte, the PDA@3DVAG composite electrode delivered 96.7 F g-1 initial specific capacitance and 61.8% capacity retention after 3000 cycles, and 46.14 Wh kg-1 energy density at the power density of 393.75 W kg-1. However, there are some important data that need to be added before it is accepted to publish.

  1. The surface area and pore size distribution of the PDA@3DVAG electrode need to be measured.
  2. The self-discharge experiment needs to be conducted, the leak current needs to be measured.
  3. In Table 1, the authors claimed they demonstrated a 1.6 V supercapacitor in this work. However, based on the experiment data, they only charged to 0.8V (Figure 7B and D). Meanwhile, most of the discharge capacity was obtained at a negative voltage. That means this supercapacitor cannot supply driven force for the external circuit during discharge.

Reviewer 3 Report

This paper is proposing a three-dimensional vertical graphene-PDA composite as hybrid super capacitor electrode. Results are interesting but the following points need to be clarified by the authors:

1) How the authors compare their results with this paper (or similar): Polydopamine-Modified Reduced Graphene Oxides as a Capable Electrode for High-Performance Supercapacitor. ChemistrySelect. 4. 2711-2715. 10.1002/slct.201900242. In general, the authors should include references that are using similar materials and compare their results. 

2) Lines 189-190: it is not clear whether the electrochemical tests were performed after 24 h of impregnation. If not, it would be useful to manufacture cells after 24 h of impregnation in order to support the statement of Figure 8 (lines 233-234). 

3) The authors should indicate: how the measure the capacitance (Galvanostatic, CV, other) and which mass of the electrode they are using to determine the F/g ratio.   

4) Figure 6(b): the legend has to be corrected (24 h instead of 24H) and also 281 nm instead of 281nm.

5) Lines 220-221: the authors claim that the low frequency range higher slope indicates lower resistance to ion transport. However, in the cases that they refer to, the slope is well higher than 1, which indicates double layer capacitance and not diffusion. The PDA/WIS/AC follows a diffusion pattern. Therefore there is a fundamental difference between these systems. In addition, data at high frequency range are not shown or commented. The value of the resistance at high frequency would reveal the difference in the ionic conductivity of the various  electrolytes. The authors are encouraged to review this part of the discussion. 

Round 2

Reviewer 1 Report

As the authors have answered the questions raised in the report, in my opinion now it can be accepted for publication in nanomaterials.

Reviewer 2 Report

The authors have made significant improvements. I think the manuscript can be accepted for publication in Nanomaterials.

Reviewer 3 Report

I am satisfied with the modifications. This paper can be published. 
